# Reduced IQGAP2 Promotes Bladder Cancer through Regulation of MAPK/ERK Pathway and Cytokines

**DOI:** 10.3390/ijms232113508

**Published:** 2022-11-04

**Authors:** Fei Song, Roland Kotolloshi, Mieczyslaw Gajda, Martin Hölzer, Marc-Oliver Grimm, Daniel Steinbach

**Affiliations:** 1Department of Urology, Jena University Hospital, 07740 Jena, Germany; 2Section of Pathology, Department of Forensic Medicine, Jena University Hospital, 07747 Jena, Germany; 3RNA Bioinformatics and High-Throughput Analysis, Friedrich Schiller University Jena, 07743 Jena, Germany

**Keywords:** bladder cancer, IQGAP2, MEK/ERK signaling, cancer progression, tumor suppressor

## Abstract

The progression of non-muscle-invasive bladder cancer (NMIBC) to muscle-invasive bladder cancer (MIBC) is a major challenge in urologic oncology. However, understanding of the molecular processes remains limited. The dysregulation of IQGAP2 is becoming increasingly evident in most tumor entities, and it plays a role in multiple oncogenic pathways, so we evaluated the role of IQGAP2 in bladder cancer. IQGAP2 was downregulated in tumors compared with normal urothelium tissues and cells. IQGAP2 effectively attenuated bladder cancer cell growth independently from apoptosis. Reduced IQGAP2 promoted EMT in bladder cancer cells via activation of the MAPK/ERK pathway. In addition, IQGAP2 might influence key cellular processes, such as proliferation and metastasis, through the regulation of cytokines. In conclusion, we suggest that IQGAP2 plays a tumor-suppressing role in bladder cancer, possibly via inhibiting the MAPK/ERK pathway and reducing cytokines.

## 1. Introduction

Bladder cancer is among the top ten diagnosed cancers, with approximately 600,000 new cases and 20,000 deaths per year worldwide [1]. At first diagnosis, 80% of the tumors are non-muscle-invasive bladder carcinomas (NMIBCs), and the recurrence rate is about 47%. About 10% to 15% of NMIBCs progress to muscle-invasive tumors (MIBCs), which are associated with a significantly poorer prognosis, metastasis, and a 5-year survival rate <50% [2,3]. Therefore, the identification of novel potential targets regulating bladder cancer initiation and progression is needed. IQGAP2, a member of the IQ motif containing GTPase-activating proteins (IQGAPs), was first identified in 1996 and is mostly expressed in the liver, gastrointestinal tract, testes, thyroid, kidney, platelets, and breasts [4]. There are three IQGAP paralogs in humans, IQGAP1, IQGAP2, and IQGAP3, sharing similar domain structures and having high sequence homology [5]. These scaffold proteins are mostly located between the cell junctions of epithelial cells and interact with components of the cytoskeleton, cell adhesion molecules, and diverse signaling molecules to regulate multiple cellular processes [6]. While major studies have focused on IQGAP1, indicating an oncogenic role in most cancers [7,8,9], IQGAP2 has received less attention and has been described as a tumor suppressor in most cancers [10,11,12,13,14], including gastric cancer, prostate cancer, ovarian cancer, and breast cancer. The reduced expression of IQGAP2 in cancer cells is correlated with cell proliferation [10,11], migration [10,12,13], invasion [10,11,12,13,14], apoptosis [10,15], angiogenesis [16], and clinical characteristics [13,17].

The function of the other two IQGAPs in bladder cancer have already been reported. IQGAP1 was reported as a tumor suppressor in bladder cancer, playing a completely opposite role compared with other cancers [18]. Expression levels of IQGAP3 in tissue and urine samples from bladder cancer patients are significantly higher than those in samples from control groups [19,20], and IQGAP3 was upregulated by CDC42 activation of the Ras/ERK pathway, promoting the development of bladder cancer [20]. However, the role of IQGAP2 in bladder cancer is still unclear.

In this study, we prove the tumor-suppressing role and identify the molecular mechanism of IQGAP2 in bladder cancer. These results suggest that IQGAP2 might have a potential role in the regulation of the progression of bladder cancer.

## 2. Results

### 2.1. IQGAP2 Expression Is Reduced in Bladder Cancer

To investigate the potential role of IQGAP2 in bladder cancer patients, we first analyzed the mRNA expression level of IQGAP2 using the Oncomine database [21]. IQGAP2 was differentially expressed in tumors compared to corresponding normal tissues. In the majority of tumor entities, as well as in bladder cancer, IQGAP2 was expressed significantly lower in tumor tissue (Figure 1A and Appendix A). In addition, when analyzing the mRNA expression of IQGAP2 with Timer using the TCGA dataset, the outcome showed a significantly reduced expression of IQGAP2 in tumors (n = 408) compared to normal bladder tissue (n = 19), as well as reduced expression compared to most other tumor entities (Figure 1B). 

We also analyzed the protein expressions of IQGAP2 in the FFPE tumor tissue samples of 11 MIBC cases and 7 matched NMIBC cases through immunohistochemistry (Appendix A). Overall, IQGAP2 seemed to be very weakly expressed in the urothelium, as well as in bladder cancer cells. However, immune cells were predominantly weakly stained. In all cases, the expressions of IQGAP2 were localized in the cytosol, not in the nucleus. In non-dysplastic urothelium, only umbrella cells were predominantly weakly stained. In dysplastic urothelium, the expression was slightly increased and concerned part-to-all of the urothelium cell layers. Non-muscle-invasive tumors, as well as muscle-invasive tumor areas, were mainly negatively or weakly stained. Interestingly, in muscle-invasive tumors, the expression varied more between tumor cell areas and samples. Here, in two cases, moderately stained tumor cell areas or a few strongly stained single tumor cells were found (patients 2, 5, and 6). On average, no significant differences were found in different stages of urothelial dysplasia or cancer (Appendix A). However, IQGAP2 was generally low-expressed in tumor samples, which is expected of a tumor-suppressing entity.

Then, we checked the IQGAP2 mRNA expression in urinary tract cancer cell lines using the Cancer Cell Line Encyclopedia (CCLE) [22]. The mean expression was lower compared to the mean expressions of cell lines from other tumor entities (Figure 1C). 

Lastly, we examined the expressions of IQGAP2 mRNA and protein in one normal urothelial cell line and four bladder cancer cell lines of T24, Cal29, TCCSup, and RT4 through our own analysis. The qPCR results indicate that bladder cancer cells had a significantly lower amount of IQGAP2 mRNA than the normal urothelial cell line of SV-HUC-1 (Figure 1D). As expected, the IQGAP2 protein was relatively higher-expressed in SV-HUC-1 compared with bladder cancer cell lines. Additionally, IQGAP2 was lost in RT4 (Figure 1E).

### 2.2. Altered IQGAP2 Expression Affects Growth of Bladder Cancer Cell Lines

We investigated whether the expression of IQGAP2 was associated with the cell growth activity of bladder cancer cells. Previous research has indicated the tumor-suppressing role of IQGAP2 in cancers, so we selected T24 and TCCSUP cells with high IQGAP2 expression for siRNA transfection to construct knockdown cell models. We also transfected plasmids containing the full CDS of IQGAP2 in T24, which had higher proliferal, migrassive, and invasive abilities, to construct an overexpression model. Knockdown (Figure 2A,D) and overexpression (Figure 2G) were confirmed by Western blot analysis. Colony formation experiments revealed that silencing IQGAP2 increased cell proliferation in T24 (2.07-fold; *p* < 0.001) and TCCSUP (1.96-fold; *p* < 0.01) cells (Figure 2B,E), whereas the ectopic expression of IQGAP2 in T24 decreased cell proliferation (0.56-fold; *p* < 0.01) (Figure 2H). Similar results were seen in the spheroid model, which, in comparison to the monolayer model, more closely reflected tumor growth in vivo in terms of complexity. The knockdown of IQGAP2 increased the spheroid volume of T24 (1.77-fold; *p* < 0.01) and TCCSUP (1.94-fold; *p* < 0.05) cells (Figure 2C,F), while the ectopic expression of IQGAP2 reduced the spheroid volume of T24 (0.54-fold; *p* < 0.05) (Figure 2I). These findings indicate that IQGAP2 suppressed bladder cancer cell proliferation and tumor growth.

### 2.3. IQGAP2 Expression Does Not Affect Apoptosis in Bladder Cancer Cells

IQGAP1 has been implicated in apoptosis in a variety of malignancies. The upregulation of IQGAP2-induced apoptosis has also been observed in breast cancer and gastric cancer. To determine if the growth inhibition effect of IQGAP2 on bladder cancer cells was also a result of apoptosis, we stained cells with PI and Annexin V and performed a FACS analysis. No significant change was observed (Figure 3A). Additionally, we examined the expressions of apoptosis markers, such as Caspase3 and BCL-2, but found no significant change after the overexpression of IQGAP2 (Figure 3B).

### 2.4. Altered IQGAP2 Expression Affects Migration, Invasion, and EMT of Bladder Cancer Cell Lines

We next used an xCELLigence RTCA System to monitor the effect of IQGAP2 on the migration and invasion activities of bladder cancer cells in real time. The knockdown of IQGAP2 resulted in a higher rate of migration (*p*-value < 0.05 starting from 59 h) and invasion (*p*-value < 0.05 starting from 52 h) properties in T24 cells (Figure 4A). Similarly, promoted migration (*p*-value < 0.05 starting from 63 h) and invasion (*p*-value < 0.05 starting from 39 h) were observed in TCCSUP after the knockdown of IQGAP2 (Figure 4B). On the other hand, ectopic expression of IQGAP2 significantly inhibited the migration (*p*-value < 0.05 starting from 16 h) and invasion (*p*-value < 0.05 starting from 8 h) of T24 (Figure 4C).

The loss of E-cadherin and increase in N-cadherin, which is called the “cadherin switch”, has long been regarded as a major hallmark of epithelial–mesenchymal transition (EMT), the primary cause of metastasis [23]. MMPs are well-known to be associated with the development of invasive bladder cancer [24,25,26]. We detected a reduced transcription level of E-cadherin (CDH1) and increased transcription levels of MMP9 and N-cadherin (CDH2) after IQGAP2 depletion in both TCCSUP (Figure 5A) and T24 cells (Figure 5B). Moreover, the overexpression of IQGAP2 in T24 cells showed the opposite effect (Figure 5C).

### 2.5. Knockdown of IQGAP2 Promotes Invasion via Activation of ERK Pathway

Following the above, we sought to determine how IQGAP2 modulated bladder cancer cell function. The c-BioPortal database was used to collect related genes to IQGAP2 regarding mRNA expression (Appendix A). The roles and pathways of these genes, as well as those of IQGAP2, were then predicted using KOBAS. The results showed an enrichment of pathways in cancer of, inter alia, Ras/MAPK and PI3K/AKT signaling pathways, and cytokine interactions were involved (Figure 6A). We first focused on examining the MAPK/ERK and PI3K/AKT signaling pathways in bladder cancer cells following knockdown or overexpression. After IQGAP2 knockdown, we observed an increase in phospho-MEK1/2 and phospho-ERK1/2 levels in T24 and TCCSUP (Figure 6B,C), whereas overexpression decreased phospho-MEK1/2 and phospho-ERK1/2 levels in T24 (Figure 6D). Additionally, silencing IQGAP2 in EX_IQGAP2 T24 cells restored the lower levels of phospho-MEK1/2 and phospho-ERK induced by IQGAP2 overexpression (Figure 6E). There were no significant findings for the AKT pathway (Appendix A). As the ERK pathway has been implicated in EMT in different cancer types, we also evaluated this in the bladder cancer cell line T24. U0126, an ERK inhibitor, effectively suppressed the IQGAP2-depletion-induced increase in phospho-ERK (Figure 6F), while also abolishing invasiveness (*p*-value < 0.05 for the time frame of 22–42 h) (Figure 6G). These findings support the hypothesis that decreased IQGAP2 expression caused invasion with the involvement of the MEK/ERK pathway.

### 2.6. Knockdown of IQGAP2 Induces Expression of Cytokines

Then, we determined the degrees of expression of pro-inflammatory cytokines, which play critical roles in tumor genesis, development, and metastasis [27]. These effects were also confirmed in bladder cancer [28]. We observed a significant increase in IL-6 and CCL2 levels in T24 and TCCSUP cells following IQGAP2 knockdown (Figure 7A,B). In contrast, the overexpression of IQGAP2 reduced the levels of IL-6, IL-8, and CCL2 transcripts (Figure 7C).

## 3. Discussion

The progression of bladder cancer from NMIBC to MIBC is a critical event during the entire course of the disease and has a substantial impact on treatment [29]. It remains difficult to predict which patients may develop from superficial to muscle-invasive disease. This is most likely related to the heterogeneous genotype and phenotype of urothelium carcinoma, which is why it is critical to investigate the mechanism of bladder cancer progression. In this study, we detected IQGAP2 expression in MIBC and early NMIBC samples from the same patients and then revealed the role of IQGAP2, a known tumor suppressor, in bladder cancer progression.

The results from public databases indicated that the amount of IQGAP2 mRNA was significantly reduced in most cancers, including bladder cancer. The loss of IQGAP2 protein has been observed in many cancer types and is linked to prognosis, including breast, gastric, and ovarian cancers [13,14,15]. Although our IHC data did not show significant reductions due to the small sample size, it displayed more heterogeneous expression in MIBC and confirmed mainly low protein levels in bladder cancer tissues. However, we are currently collecting more bladder cancer samples of different stages to confirm the role of IGQAP2 as a prospective marker in bladder cancer.

Our in vitro data clearly demonstrated that IQGAP2 had an antiproliferative effect in bladder cancer cells, which is comparable with previous findings in other cancer types [10,11,12,13,14]. Similar results were obtained using the spheroid model, which more closely resembled tumor growth in vivo than a monolayer model in terms of complexity. The evidence presented above substantiated IQGAP2’s role in suppressing cell proliferation, which was independent from apoptosis.

The invasion of cancer cells into the bladder muscle layer is the primary criterion for bladder cancer staging and is a significant predictor of prognosis and survival for bladder cancer patients [30]. As previously demonstrated in numerous studies, IQGAP2 can suppress the metastasis of several types of cancer [10,11,12,13,14]. While silencing IQGAP2 facilitated the migration and invasion of bladder cancer cell lines, ectopic IQGAP2 had the reverse effect. Further investigation of the EMT markers E-cadherin and MMP9 indicated decreased IQGAP2-induced EMT in bladder cancer cells.

We next identified the network for IQGAP2 and its co-expressed genes to better examine its intracellular function. The KEGG pathway analysis revealed that the Ras/MAPK and PI3K/AKT signaling pathways were concerned, which have been linked to migration and invasion in bladder cancer [31,32]. In our investigation, silencing IQGAP2 triggered MEK/ERK signaling, resulting in an invasive phenotype. Previous reports have indicated that Ras, which regulates various cellular functions by cycling between GTP-bound active and GDP-bound inactive states, shows different binding affinities to IQGAPs. IQGAP2 has been observed to interact without discrimination with both the GDP and GTP-bound versions [4,33]. Additionally, direct physical connection between the polyproline-binding region (WW) and ERK 1/2, as well as between the IQ motif (IQM) and MEK1/2, has been reported [8,9]. However, only Kumar, D. et al. reported a direct effect of IQGAP2 level on ERK binding [15]. Therefore, the mechanism by which IQGAP2 regulates the MEK/ERK pathway requires more in-depth study. 

Apart from the MAPK pathway, the KEGG analysis revealed a high link between IQGAP2 and cytokines. In this investigation, we found that silencing IQGAP2 increased IL-6 and CCL2 levels in bladder cancer cells, whereas the overexpression of IQGAP2 lowered IL-6, IL-8, and CCL2 levels. According to previous research, these cytokines are upregulated in bladder cancer and are associated with proliferation, migration, invasion, and angiogenesis [34,35,36,37,38]. We suggest that cytokine expression could be suppressed by IQGAP2 in bladder cancer cells. The present results corroborate prior observations, providing insight into how IQGAP2 may influence critical cellular processes, such as proliferation and metastasis, via cytokine modulation. Indications from other tumor entities have proved that wnt signaling is also an important pathway involved in IQGAP2′s functional role [13,39]. Nevertheless, we detected no significant effect on wnt target gene expression after the knockdown of IQGAP2 (Appendix A). However, we showed that IQGAP2 acted as a tumor suppressor by the inhibition of growth, migration, and invasion in bladder cancer cells, probably through the regulation of the MAPK/ERK pathway and cytokines, although it is not known how IQGAP2 is regulated. This topic needs to be the subject of further studies.

## 4. Materials and Methods

### 4.1. Patients and Samples

Detailed methodology is given as described previously [40]. Tumor specimens were collected from bladder cancer patients at the Department of Urology, Jena University Hospital, Germany. The clinical characteristics (age, sex, tumor stage, and grade) of the patients are summarized in Table 1. All patients provided written informed consent to donate leftover tissue for research purposes and storage in the institutional biobank. Jena University Hospital’s institutional ethical committee approved the biobank (No. 3657-01/13, confirmed on 7 January 2020) and the submitted work (No. 2019-1569-Material). The study was conducted in accordance with the Declaration of Helsinki.

### 4.2. Cell Lines

T24, TCCSUP, and RT4 cell lines were cultured in RPMI 1640 medium (Thermo Scientific, Waltham, MA, USA); the Cal29 cell line was cultured in DMEM (Thermo Scientific, USA); and SV-HUC-1 was cultured in F12K (Thermo Scientific, USA). All media were supplemented with 10% FBS, penicillin (100 U/mL), and streptomycin (100 µg/mL). All the bladder cancer cell lines were commercially obtained in the 1990s, and short tandem repeat (STR) analysis was performed to confirm the genetic profiles of the cell lines. The normal urothelium cell line of SV-HUC-1 was purchased from amsbio, Germany. All cells were incubated in a humidified atmosphere at 37 °C with 5% CO_2_.

### 4.3. Immunohistochemistry

Sections of 4 µm from formalin-fixed paraffin-embedded (FFPE) tumor samples were immunohistochemically stained with Dako REAL EnVision Detection System K5007 (Agilent, Germany) according to the manufacturer’ s protocol against IQGAP2 (anti-IQGAP2 1:50, sc-17835, Santa Cruz, CA, USA). Briefly, tumor samples underwent deparaffinization, and heat-induced epitope retrieval was performed in sodium citrate buffer (10 mM sodium citrate, 0.05% Tween 20, pH 6.0) at 98 °C for 20 min. A consecutive section was stained with hematoxylin–eosin. Immunohistochemical staining was assessed by intensity (no staining or negative, weak, moderate, or strong) of the tumor cells.

### 4.4. siRNA, Expression Vector, and Transfection

For knockdown, cells were transfected with siGENOME Human IQGAP2 siRNA SMARTPool (M-006457-02-0005) or siGENOME Non-Targeting siRNA #1 (D-001210-01-20) (Horizon, Cambridge, UK) using INTERFERin reagent following the manufacturer’s protocol (Polyplus, New York, NY, USA). For overexpression, cells were transfected with IQGAP2_OHu14310C_pcDNA3.1(+) (NM_006633.4) or empty pcDNA3.1(+) (Genescript, Piscataway, NJ, USA) using Jetprim (Polyplus, USA). For stable expression, cells were grown in complete media supplemented with G418 (300 µg/mL) antibiotics.

### 4.5. RNA Extraction and Quantitative Real-Time PCR (qRT-PCR)

Total RNA from cultured cells was extracted using TRIzol reagent (Thermo Fisher Scientific, USA) according to the manufacturer’s instructions. RNA was quantified using a Qubit 3.0 Fluorometer (Thermo Scientific, Waltham, MA, USA), and RNA quality was analyzed with an Agilent 2200 TapeStation (Agilent, Santa Clara, CA, USA) instrument according to the manufacturers’ protocols. Total RNA was then reverse-transcribed into cDNA using a GoScript Reverse Transcription System (Promega, Walldorf, Germany) according to the manufacturer’s protocol. RT-qPCR analysis was performed using LightCycler 480 SYBR Green I master mix and a LightCycler 480 instrument (Roche, Munich, Germany). Sequences of primers are provided in Appendix A. RPS23 was used as the reference control, and PCR was performed in triplicate, at least.

### 4.6. Western Blotting

Cells were rinsed three times with cold DPBS, transferred into a Bioruptor tube for sonication (Diagenode SA, Seraing, Belgium), and centrifuged at 1500 rpm for 10 min at 4 degrees to aspirate the supernatant. RIPA buffer supplemented with EDTA-free Protease Inhibitor Cocktail (Roche, Germany) and PhosSTOP (Roche, Germany) was added, and cells were incubated on ice for 20 min. Then, the tubes proceeded to sonication using a Bioruptor Pico according to the manufacturer’s protocol. The cell debris was removed by centrifugation (15,000 rpm, 15 min, 4 °C). The protein extracts were separated and blotted on a polyvinylidene difluoride (PVDF) membrane with 4–12% SDS-PAGE (Serva, Heidelberg, Germany). The following antibodies were used: anti-IQGAP2 (sc-17835), anti-p-ERK1/2 (sc-7383), anti-p-MEK1/2 (sc-81503), anti-p-AKT/1/2/3 (sc-514032), anti-GAPDH (sc-47724), and anti-α-Tubulin (sc-5286) (Santa Cruz Biotechnology, Dallas, TX, USA). Horseradish-peroxidase-conjugated anti-mouse IgG-HRP (sc-516102, Santa Cruz Biotechnology, USA) was used as a secondary antibody. Membranes were incubated with chemiluminescence (ECL) reagent (GE Healthcare, Solingen, Germany), and the produced light emission was detected with a GBox Chemi XX6 (Syngene, Cambridge, UK) instrument. Analyses were performed in triplicate, at least.

### 4.7. Growth and Spheroid Formation Assays

Cells were plated on six-well culture plates and cultured at 37 °C in 5% CO_2_. After 6 days, cells were fixed with 4% formaldehyde for 30 min and stained with crystal violet (0.1% *w*/*v* in ddH_2_O). The excess dye was carefully washed away with ddH_2_O. The stained cells were lysed with 0.1% acetic acid. The absorbance was measured at 590 nm (Tecan, Männedorf, Switzerland).

For spheroid formation, 1000 cells were seeded per well of a cell-repellent surface 96-well microplate (Greiner Bio-one, Frickenhausen, Germany). Afterward, the plate was centrifuged at 800 rpm for 3 min. After 6 days of incubation, spheroids were observed under a microscope, and pictures were captured with a digital camera (Canon G16). MATLAB R20019a and AnaSP software were used to extract the volume from spheroid formation experiments. All experiments for proliferation and spheroid formation were performed in triplicate, at least.

### 4.8. xCELLigence Real-Time Cellular Analysis

Migration and invasion of cells were analyzed using an xCELLigence RTCA System (ACEA Bioscience, San Diego, CA, USA) according to the manufacturer’s protocol. For migration, 20,000 cells in 100 μL were added to the upper chamber of CIM plates. In the case of invasion, 40,000 cells in 100 μL were added to upper chamber wells pre-coated with diluted Matrigel (catalogue no. 354230, Corning, New York, NY, USA). In the lower chamber, medium with 10% serum was added. All experiments were repeated at least in triplicate. For statistical analysis, the means of three independent biologicals were combined, and a two-tailed Student’s *t*-test was performed at any point in time of measurement.

### 4.9. Cell Apoptosis Assay

Apoptosis was detected with an FITC Annexin V apoptosis detection kit (556547, BD Pharmingen, Franklin Lakes, NJ, USA) following the manufacturer’s instruction, which included dual staining with annexin V-FITC and propidium iodide (PI), and were analyzed with a flow cytometry BD Accuri C6 Plus (BD Pharmingen) instrument. Data were analyzed using BD Accuri C6 Plus software.

### 4.10. Analysis of Expression Level of IQGAP2 in Pan-Cancer Datasets

Oncomine is a cancer microarray database and web-based data-mining platform that facilitates fast interpretation of a gene’s probable role in cancer and aids in genome-wide expression investigations [21]. We used the Oncomine database [https://www.oncomine.org, Compendia biosciences, Ann Arbor, MI, USA] to compare the expressions of IQGAP2 in various subtypes of cancer on 2 December 2021. Timer is a tool that allows users to study the differential expressions between tumors and adjacent normal tissues for any gene of interest across all TCGA tumors [41]. We studied IQGAP2 mRNA expressions in different TCGA tumors using Timer on 1 December 2021. Cancer Cell Line Encyclopedia (CCLE) (https://portals.broadinstitute.org/ccle) includes the large-scale in-depth sequencing of 947 human cancer cell lines, covering more than 30 tissue sources and integrating genetic information, such as DNA mutations, gene expressions, and chromosome copy numbers [22]. IQGAP2 expressions in human cancer cell lines were obtained from the CCLE dataset on 1 December 2021.

### 4.11. Functional Annotation and Pathway Mapping

c-BioPortal (https://www.cbioportal.org/) was used to collect the mRNA expression data of bladder urothelial carcinoma (408 patients), as well as the TCGA (cell 2017) dataset [23] for IQGAP2 and its frequently altered correlated genes (Spearman’s correlation > 0.5), on 29 November 2021. KOBAS 3.0 [42] (http://kobas.cbi.pku.edu.cn/), a web-based tool for gene and protein functional annotation and pathway enrichment analysis, was used to predict the enriched pathways (filter: *p* < 0.001) of the these changed genes on 29 November 2021.

### 4.12. Data and Statistical Analysis

All statistical analyses were conducted using GraphPad Prism 8.0 and Microsoft Excel. Continuous data were presented as means ± standard error of means and evaluated using Student’s *t*-test (two-tailed, unpaired). For all tests, *p* < 0.05 was considered significant (* *p* ≤ 0.05, ** *p* ≤ 0.01, and *** *p* ≤ 0.001).

## 5. Conclusions

This study initially showed that IQGAP2 was reduced in bladder cancer cells. Moreover, an in vitro analysis indicated that IQGAP2 acted as a tumor suppressor in bladder cancer through the inhibition of tumor growth, migration, and invasiveness. The MAPK/ERK pathway and cytokines were probably involved in this functional modulation. Thus, these findings implied that IQGAP2 may be a promising target for reversing the clinical progression of NMIBC to MIBC.

## Figures and Tables

**Figure 1 ijms-23-13508-f001:**
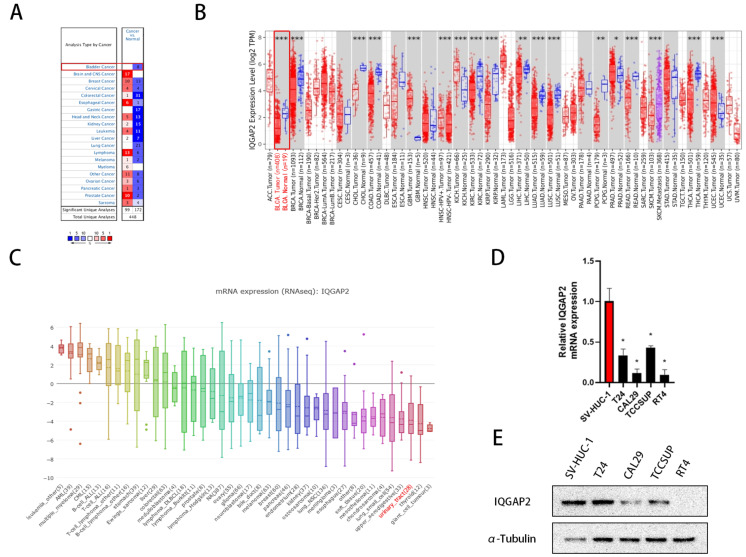
Expressions of IQGAP2 in bladder cancer samples and cell lines. (**A**) The transcription level of IQGAP2 was significantly decreased in normal tissue compared to tumor tissue in different types of cancers, including bladder cancer (Oncomine database). (**B**) The expression levels of IQGAP2 in TCGA tumor tissues and their corresponding normal tissues showed reduced expressions in bladder cancer (Timer). (**C**) The expressions of IQGAP2 in cancer cell lines derived from the urinary tract were low compared to other tumor entities (CCLE) (* *p* < 0.05). (**D**) The mRNA levels of IQGAP2 in bladder cancer cell lines were decreased compared to urothelial cell line SV-HUC-1. Bar chart presents the means of replicates ± SEM from three different experiments; * *p*  ≤  0.05, ** *p* ≤ 0.01, and *** *p* ≤ 0.001; Student’s *t*-test (two-tailed, unpaired) was used for comparison of means. (**E**) The protein levels of IQGAP2 were also decreased in bladder cancer cell lines compared to the urothelial cell line SV-HUC-1 and was lost in T24.

**Figure 2 ijms-23-13508-f002:**
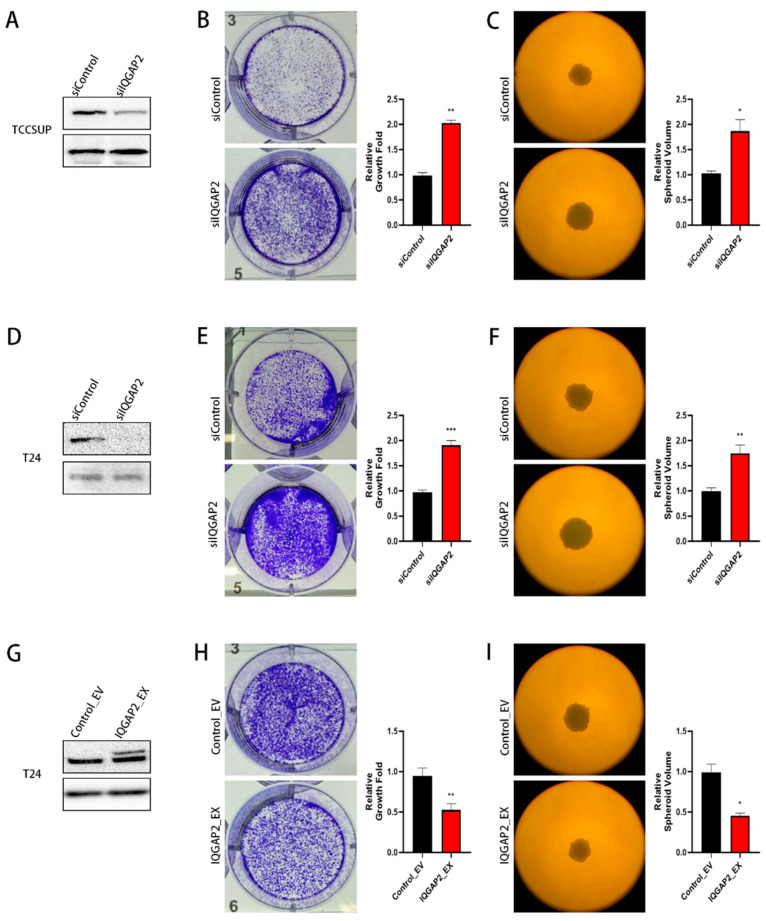
Reduced IQGAP2 promoted cell growth in bladder cancer cell lines. IQGAP2 knockdown in TCCSUP (**A**) and T24 (**D**) was confirmed by Western blot analysis. Knockdown of IQGAP2 in TCCSUP (**B**,**C**) and T24 (**E**,**F**) led to significant increase in proliferation in cell monolayer and spheroid growth. IQGAP2 overexpression in T24 (**G**) was confirmed by Western blot analysis. Overexpression of IQGAP2 in T24 (**H**,**I**) led to significant decrease in proliferation in cell monolayer and spheroid growth. One example in triplicate is shown; bar chart presents the means of replicates ± SEM from three different experiments; * *p* ≤ 0.05, ** *p* ≤ 0.01, and *** *p* ≤ 0.001; Student’s *t*-test (two-tailed, unpaired) was used for comparison of means.

**Figure 3 ijms-23-13508-f003:**
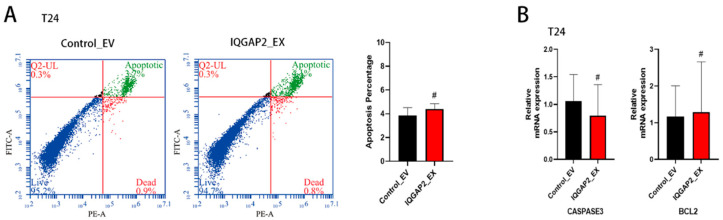
IQGAP2 did not influence cell apoptosis in bladder cancer cells. (**A**) Cell apoptosis was measured using flow cytometry in IQGAP-overexpressing (IQGAP_EX) T24 and control cells (empty vector—EV). (**B**) Cell apoptosis markers Caspase 3 and BCL2 showed no different mRNA expression in T24 IQGAP2_EX and Control_EV. One example of replicates is shown; bar chart presents the means of replicates ± SEM from three different experiments; # indicates *p* > 0.05; Student’s *t*-test (two-tailed, unpaired) was used for comparison of means.

**Figure 4 ijms-23-13508-f004:**
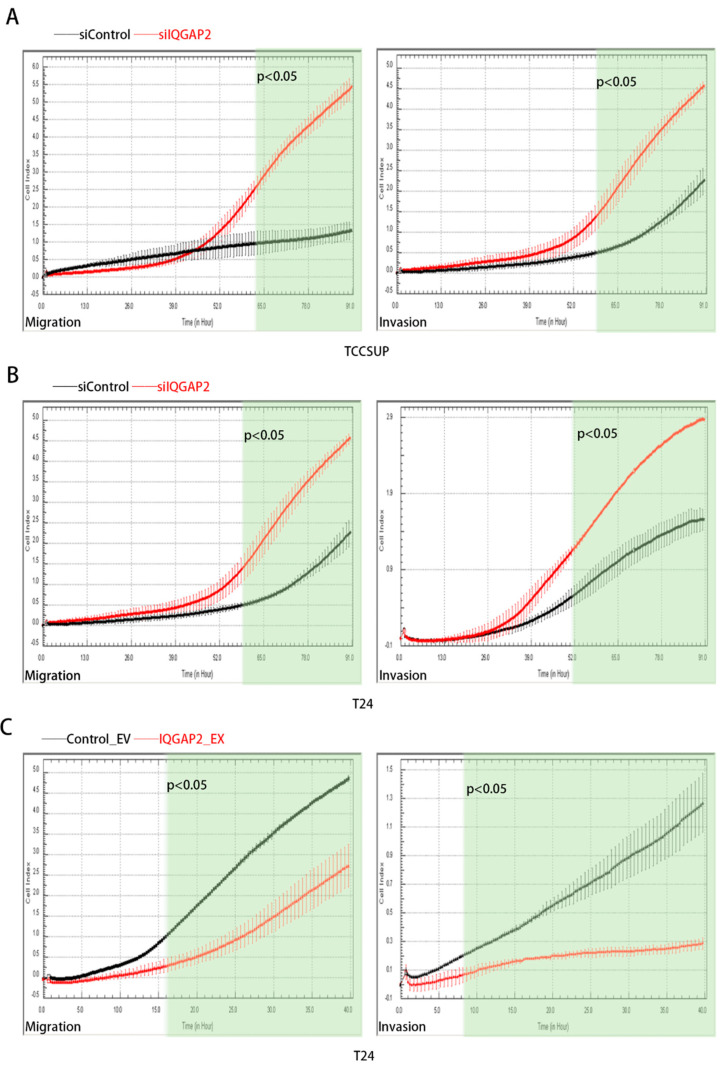
IQGAP2 influenced migration and invasion of bladder cancer cells. (**A**) Migration and invasion were significantly increased after knockdown of IQGAP2 in TCCSUP. (**B**) Migration and invasion were significantly increased after knockdown of IQGAP2 in T24. (**C**) Migration and invasion were significantly decreased after overexpression of IQGAP2 in T24. Bar chart presents the means of replicates ± SEM from three different experiments; Student’s *t*-test (two-tailed, unpaired) was used for comparison of means.

**Figure 5 ijms-23-13508-f005:**
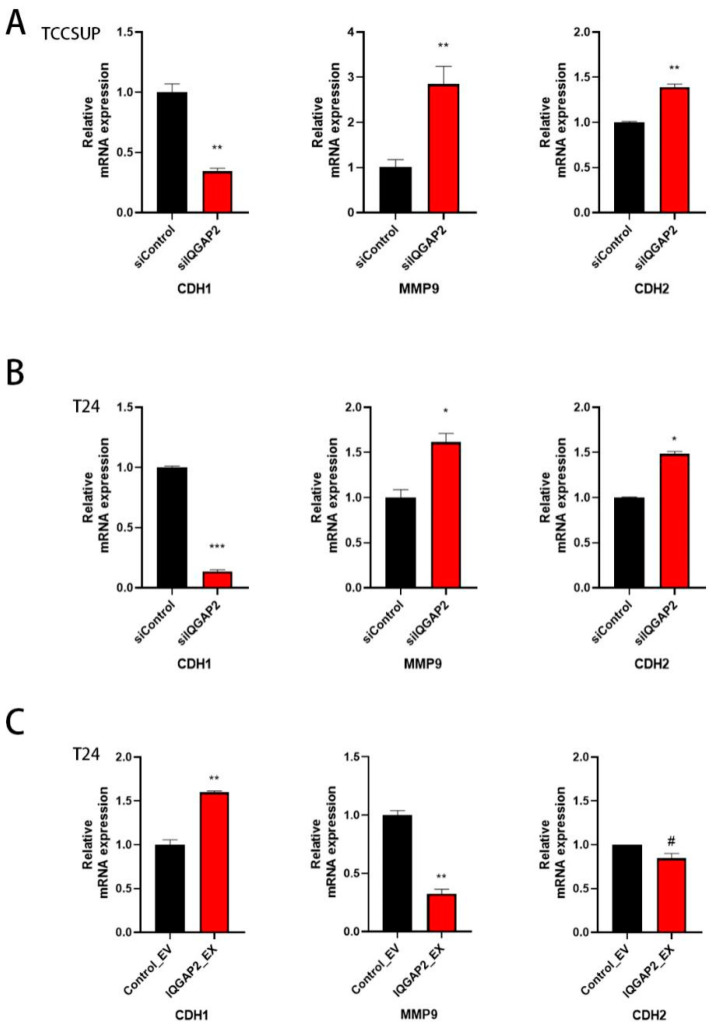
IQGAP2 expression levels showed effects on EMT markers. qPCR showed elevated expressions of MMP9 and CDH2 and a reduced expression of CDH1 in TCCSUP (**A**) and T24 (**B**) with IQGAP2 knockdown (siIQGAP2) compared to control (siControl). qPCR show reduced expression of MMP9 and elevated expression of CDH1 in T24 (**C**) with IQGAP2 overexpression (IQGAP2_EX) compared to control (Control_EV). Bar chart presents the means of replicates ± SEM from three different experiments; Student’s *t*-test (two-tailed, unpaired) was used for comparison of means; # indicates *p* > 0.05; * *p* ≤ 0.05, ** *p* ≤ 0.01, and *** *p* ≤ 0.001.

**Figure 6 ijms-23-13508-f006:**
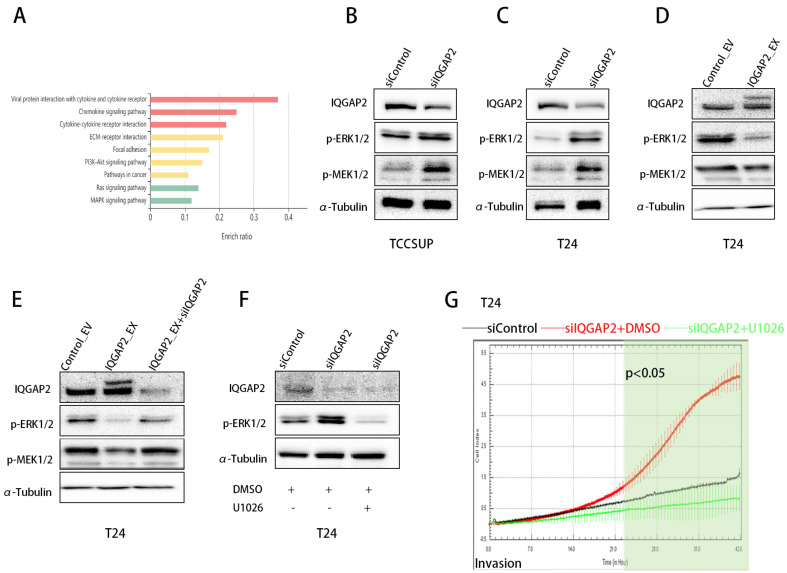
IQGAP2 regulated EMT via activation of MAPK/ERK pathway. (**A**) KEGG pathway analysis (http://kobas.cbi.pku.edu.cn/ accessed on 29 November 2021) of IQGAP2 and its co-expressed genes in bladder cancer identified by cBioPortal co-expression analysis using bladder cancer TCGA dataset (www.cbioportal.org accessed on 29 November 2021). (**B**,**C**) Western blot showing elevated phospho-MEK1/2 and phospho-ERK1/2 in TCCSUP and T24 after IQGAP2 knockdown. (**D**) Western blot showing decreased phospho-MEK1/2 and phospho-ERK1/2 in T24 after IQGAP2 overexpression. (**E**) Western blot showing rescue of phospho-MEK1/2 and phospho-ERK1/2 in T24_IQGAP2_EX with IQGAP2 depletion compared with control. (**F**) Western blot showing rescued phospho-ERK after treating IQGAP2-depleted (IQGAP2_KD) T24 with phospho-ERK inhibitor U0126. (**G**) Cell invasion assay after treating IQGAP2-depleted (IQGAP2_KD) T24 with phospho-ERK inhibitor U0126. Bar chart presents the means of replicates ± SEM from three different experiments; Student’s *t*-test (two-tailed, unpaired) was used for comparison of means.

**Figure 7 ijms-23-13508-f007:**
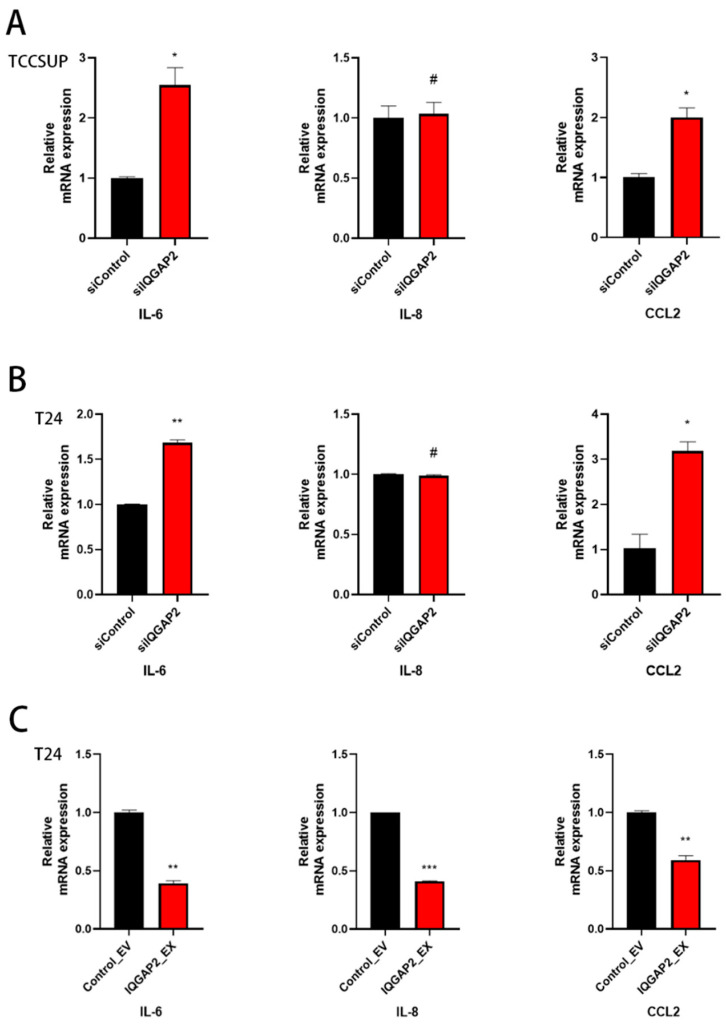
IQGAP2 expression reduced pro-inflammatory cytokine expressions in bladder cancer cells. qPCR showed elevated expressions of IL6 and CCL2 in TCCSUP (**A**) and T24 (**B**) with IQGAP2 knockdown (siIQGAP2) compared to control (siControl). qPCR showed reduced expressions of IL6, IL8, and CCL2 in T24 (**C**) with IQGAP2 overexpression (IQGAP2_EX) compared to control (Control_EV). Bar chart presents the means of replicates ± SEM from three different experiments; Student’s *t*-test (two-tailed, unpaired) was used for comparison of means; # indicates *p* > 0.05; * *p* ≤ 0.05, ** *p* ≤ 0.01, and *** *p* ≤ 0.001.

**Table 1 ijms-23-13508-t001:** Patient material and characteristics. NMIBC, non-muscle-invasive bladder cancer; MIBC, muscle-invasive bladder cancer; LG, low grade; HG, high grade; m, male; f, female.

Patient			Primary NMIBC and Recurrence	PFS [m]	MIBC
No	Age	Sex	TNM-T/Grading		TNM-T/Grading
1	67	m	pTa HG, pT1 HG	15	pT4a HG
2	79	f	pT1 HG, pTa LG, 3x pTa HG	5	pT2b HG
3	69	m	pTa HG	10	pT4a HG
4	78	f	pT1 HG, pTa HG	12	pT4a HG
5	65	m	pTa HG	21	pT2a HG
6	71	m	pT1 HG, pTa LG	23	pT2a HG
7	74	m	pTa HG	42	pT3a HG
8	73	m	pTa LG	62	pT2 HG
9	63	m	pTa HG, pTa HG	10	pT2a HG
10	76	m	pTa HG	16	pT3b HG
11	76	m	pTa HG, pTa HG	28	pT4a HG

## Data Availability

Data are contained within the article or Appendix A.

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
