# Peer review of "Reduced IQGAP2 Promotes Bladder Cancer through Regulation of MAPK/ERK Pathway and Cytokines"

_ijms, 2022, doi:10.3390/ijms232113508_

Round 1
Reviewer 1 Report
Authors have found that IQGAP2 might play a tumor suppressor role in bladder cancer possibly via inhibiting MAPK/ERK pathway and reducing cytokines. It is a novel finding of IQGAP2, an IQGAP family and I believe this could be a crucial clue for further investigation of bladder cancer pathology.
However, a minor revision is needed.
1. The mutation rate in MIBC is relatively insufficient to suggest its relationship with bladder cancer.
2. In the result part (2.4.), there are 3D and E mentioned in the article, but in the picture, it is Fig. 3A and B.
3. How was the ectopic expression of IQGAP2 in TCCSUP cells, please state the explanation for the missed results in TCCSUP cells in the appropriate part.
Author Response
Dear Reviewer #1,
Thank you for reviewing our manuscript, your comments and constructive criticism. We hope that we have been able to answer all of your comments satisfactorily.
> The mutation rate in MIBC is relatively insufficient to suggest its relationship with bladder cancer.
You are right, the evidence that detected mutations in MIBC in 3 of 8 patients are associated with the disease is weak, due to low allele frequency and missing validation by re-sequencing. Nevertheless, this finding was the motivation to analyze the expression and function of IQGAP2 in bladder cancer.
To take into account your criticism we mentioned this fact in the results and the discussion part of the revised manuscript:
“The allele frequency of the mutations in patient 1 and 6 was very low and we didn’t vali-date the results by re-sequencing so far. Therefore, the evidence of the association of these mutations to the disease is not strong. Furthermore, in the Cancer Genome Atlas Study of Bladder Cancer 2017, whole-exome sequencing of 408 MIBC detected that IQGAP2 is mu-tated in only 9 patients out of 408 (2%). Nevertheless, our findings was the motivation to analyze the expression and function of IQGAP2 in bladder cancer.”
…
“In this research, we detected mutated IQGAP2 in MIBC samples, even if only with low evidence of association with the disease, and revealed the role of IQGAP2, a known tumor suppressor, in bladder cancer progression.“
> In the result part (2.4.), there are 3D and E mentioned in the article, but in the picture, it is Fig. 3A and B
Thank you for pointing out our careless mistake. We have revised the wrong number of figures.
> How was the ectopic expression of IQGAP2 in TCCSUP cells, please state the explanation for the missed results in TCCSUP cells in the appropriate part.
Thank you very much for your suggestion. Previous researches indicated the tumor suppressor role of IQGAP2 in cancers, so we selected T24 and TCCSUP cells with high IQGAP2 expression for siRNA transfection to construct knockdown cell models. Compared with TCCSUP, T24 cell line has higher proliferal, migrassive and invasive ability, so we only chose T24 which is more suitable to contrust the overexpression model for tumor suppressor. We already included this statement in the revised manuscript.
Best regards
Daniel Steinbach

Reviewer 2 Report
Re: Reduced IQGAP2 promotes bladder cancer through regulation 2 of MAPK/ERK pathway and cytokines by Song et al.
In this manuscript, the authors wanted to focus on studying the expression characteristics of gene IQGAP2 in bladder cancers. To provide a sound rationale, they managed to perform exome sequencing and discovered candidate somatic alterations in IQGAP2 in 3 out of 8 patients. They next tried to reinforce their rationale by including 408 TCGA bladder cancer patients, of which 9 had mutations in IQGAP2.
Because this reviewer has extensive experience in somatic mutation analysis, but not in expression analysis, I stopped here with a few critical questions below.
1. The description of discovered mutations is superficial and does not provide information for readers/reviewers. They listed “0.48 AF at position 910 (G>A pt 8), 0.1 AF at position 2148 (G>C, pt 1), 0.2 AF at position 4151 (G>A, pt ????)”.
2. Typically, for cancer genomics work the authors need to provide all detected mutations across all samples, including information such as chromosome, position, reference allele, mutant allele, gene name, refSeqID, protein annotation of the mutation, number of mutant allele, number of reference allele (or total depth). In this way, experienced readers/reviewers can tell whether the authors claims are likely false positives.
3. The allele fraction of 0.48, 0.1 (which barely pass their described cutoff), and 0.2 lead this reviewer to suspect their rationale and conclusion. Here, 0.1 and 0.2 would indicate only a small fraction of cells in corresponding tumor harbor this mutation. Alternatively, the low allele fraction could indicate they are possibly false positive calls. --- a validation re-sequencing would be warranted before trusting the exome data.
4. The biggest problem to this reviewer is on the patient prevalence of IQGAP2 mutation between their own population (3/8=38%) against TCGA population (9/408=2%). Such a dramatic difference indicates something is wrong.
5. Notably, the TCGA bladder paper (PMID: 28988769) discovered 58 significantly mutated genes via rigorous statistical analysis, and IQGAP2 is not in the list. On the other hand, I do not see how the authors evaluate statistical significance in their own cohort.
Overall, I do not think the authors presented any convincing data to support their rationale of hypothesizing IQGAP2 to be related to bladder cancer from cancer genomics perspective. On the other hand, I do recognize the possibility of IQGAP2 might be associated to bladder cancer from other perspectives. If the other expression-expert reviewers consider the experiments being credible, I would suggest the authors just remove this cancer genomics analysis and focus straightly on expression assay.
By the way, the authors indicated “m: male; f: female” in legend of table 1, but in the column of “Sex” I only see label m/w.
Author Response
Dear Reviewer 2,
Thank you for reviewing our manuscript, your comments and constructive criticism. We hope that we have been able to answer all of your comments satisfactorily.
- Comment: The description of discovered mutations is superficial and does not provide information for readers/reviewers. They listed “0.48 AF at position 910 (G>A pt 8), 0.1 AF at position 2148 (G>C, pt 1), 0.2 AF at position 4151 (G>A, pt ????)”.
We revised this text passage and added detailed information about the mutations, such as affected exons and amino acid changes. The missing patient of the third mutation was an error and was corrected.
- Typically, for cancer genomics work the authors need to provide all detected mutations across all samples.
You are right. Since the detected mutations of IQGAP2 in MIBC were only the motivation to analyze the expression and function of IQGAP2 in bladder cancer we did not consider it necessary to provide all detected mutations. However, the entire sequencing data of all 11 patients have been deposited in the European Nucleotide Archive under accession number PRJEB41286 (https://www.ebi.ac.uk/ena/browser/view/PRJEB41286), which is mentioned in the results part. Now, we additionally provide a excel file with the detected mutations across all samples in the supplementary materials, including genomic position, gene name, quality score, raw-depth (coverage), kind of mutation, reference and altered amino acid, etc.
- The allele fraction of 0.48, 0.1 (which barely pass their described cutoff), and 0.2 lead this reviewer to suspect their rationale and conclusion. Here, 0.1 and 0.2 would indicate only a small fraction of cells in corresponding tumor harbor this mutation. Alternatively, the low allele fraction could indicate they are possibly false positive calls. --- a validation re-sequencing would be warranted before trusting the exome data.
In our previous genomic study we sequenced the microdissected non-muscle invasive tumor and the matched metachronous muscle invasive tumor of eight patients. The low number of cases was limited by appropriate and available frozen tissue of this progressive disease. Finally, we found 8 genes with de-novo mutations (mutations exclusively in the muscle invasive tumor) in at least three patients, one of this is IQGAP2. This fact was the motivation to analyze IQGAP in detail. We know that the low allele frequency in two cases suggests that only few cells are affected and may be not related to the disease. Validation by re-sequencing was not done and must be make up.
- and 5. The biggest problem to this reviewer is on the patient prevalence of IQGAP2 mutation between their own population (3/8=38%) against TCGA population (9/408=2%). Such a dramatic difference indicates something is wrong.
Notably, the TCGA bladder paper (PMID: 28988769) discovered 58 significantly mutated genes via rigorous statistical analysis, and IQGAP2 is not in the list. On the other hand, I do not see how the authors evaluate statistical significance in their own cohort.
With the analysis of only 8 patients, we do not consider it useful to perform statistical analyzes to detect significantly mutated genes. In this scenario, any statistical significance would be very questionable. Furthermore, the TCGA study compared MIBC with normal tissue, in our study we compare non-muscle invasive tumors with muscle invasive tumors.
In conclusion, the evidence that detected mutations in MIBC in 3 of 8 patients are associated with the disease is weak, due to low allele frequency and missing validation by re-sequencing. Nevertheless, this finding was the motivation to analyze the expression and function of IQGAP2 in bladder cancer.
To take into account your criticism and comments we mentioned this fact in the results and the discussion part of the revised manuscript:
“The allele frequency of the mutations in patient 1 and 6 was very low and we didn’t vali-date the results by re-sequencing so far. Therefore, the evidence of the association of these mutations to the disease is not strong. Furthermore, in the Cancer Genome Atlas Study of Bladder Cancer 2017, whole-exome sequencing of 408 MIBC detected that IQGAP2 is mu-tated in only 9 patients out of 408 (2%). Nevertheless, our findings was the motivation to analyze the expression and function of IQGAP2 in bladder cancer.”
…
“In this research, we detected mutated IQGAP2 in MIBC samples, even if only with low evidence of association with the disease, and revealed the role of IQGAP2, a known tumor suppressor, in bladder cancer progression.“
Best regards
Daniel Steinbach

Round 2
Reviewer 2 Report
Re: Revision of “Reduced IQGAP2 promotes bladder cancer through regulation 2 of MAPK/ERK pathway and cytokines” by Song et al.
The authors have responded a few technical/presentation concerns that I have raised.
Regarding the few scientific concerns, such as the validity of the 3 alterations in their cohort of 8 cases, they stated in the revised manuscript that “Therefore, the evidence of the association of these mutations to the disease is not strong” (due to lack of validation). I am not sure how readers should integrate such information into their knowledgebase.
A critical concern that I have was the discrepancy between their frequency 3/8 vs that of TCGA frequency (9/408). Their way of dealing with discrepancy is non-scientific: “…, the evidence of the association of these mutations to the disease is not strong. Furthermore, in the Cancer Genome Atlas Study of Bladder Cancer 2017, whole-exome sequencing of 408 MIBC detected that IQGAP2 is mu-tated in only 9 patients out of 408 (2%). Nevertheless, our findings was the motivation to analyze the expression and function of IQGAP2 in bladder cancer.”
Based on above data, the scientific rigor of the authors appears to be: “whatever strong discrepancy we see, we just ignore it”.
Since I do not have expertise to judge on the remaining content, I cannot speak more on it. If the authors truly want to publish this work with high confidence on their experiments, I just suggest them to reconsider the genomic evidence they have presented.
Author Response
Round 2 Reviewer 2
Re: Revision of “Reduced IQGAP2 promotes bladder cancer through regulation 2 of MAPK/ERK pathway and cytokines” by Song et al.
The authors have responded a few technical/presentation concerns that I have raised.
Regarding the few scientific concerns, such as the validity of the 3 alterations in their cohort of 8 cases, they stated in the revised manuscript that “Therefore, the evidence of the association of these mutations to the disease is not strong” (due to lack of validation). I am not sure how readers should integrate such information into their knowledgebase.
A critical concern that I have was the discrepancy between their frequency 3/8 vs that of TCGA frequency (9/408). Their way of dealing with discrepancy is non-scientific: “…, the evidence of the association of these mutations to the disease is not strong. Furthermore, in the Cancer Genome Atlas Study of Bladder Cancer 2017, whole-exome sequencing of 408 MIBC detected that IQGAP2 is mu-tated in only 9 patients out of 408 (2%). Nevertheless, our findings was the motivation to analyze the expression and function of IQGAP2 in bladder cancer.”
Based on above data, the scientific rigor of the authors appears to be: “whatever strong discrepancy we see, we just ignore it”.
Since I do not have expertise to judge on the remaining content, I cannot speak more on it. If the authors truly want to publish this work with high confidence on their experiments, I just suggest them to reconsider the genomic evidence they have presented.
Responding:
Thank you for reviewing our manuscript, your comments and constructive criticism. We hope that we have been able to answer all of your comments satisfactorily.
We are sorry that we did not make ourselves clear from the beginning. We did observe the big discrepancy between our cohort and TCGA data, and will discuss the possible reasons more in detail in the revised manuscript.
- We are aware of the small sample size in our study, which makes any statistical tests for significance between our samples difficult. We therefore further clarified in the manuscript that the signal that we describe for IQGAP2 is not a significant result in comparison to analyzing a larger co-hort. However, our bioinformatics workflow (including GATK best practices for data preparation and LoFreq for variant calling) yielded the presented results for mutated genes (although we also experienced low allele frequencies, which we also disclaimed more clearly in the manuscript now). We therefore came up with a hypothesis and continued with explorative data analysis to identify genes with possible associations to progression of bladder cancer. Nevertheless, you’re completely right that the TCGA data indicates that IQGAP2 is not associated with bladder cancer disease. But we hope our study will motivate further genomic researches on IQGAP2 to analyze its potential connection to the disease.
- As mentioned above, the bioinformatics analyses method in our study was different to the TCGA study. We used LoFreq to call significant mutations; a sensitive variant-caller for inferring SNVs and INDELs from sequencing data via also incorporating base-call qualities and mapping/alignment uncertainties. In addition, many parameters like the general read QC, trimming of the reads, coverage cutoffs, and allele frequency cutoffs influence the mutation calls. In the TCGA study, MutSig, an algorithm used to analyze lists of mutations to identify genes that are mutated more often than expected by chance given a background mutation noise was applied. Therefore, the tool is able to detect driver mutations in high-throughput cancer genomics analysis with sufficient sample size. For our data set this was not possible. When MutSig would be applied to our much smaller sequencing data set comprising only 8 patients, we would expect no significant results. Therefore, we only took signals from our mutation calling as hints to look deeper at certain genes. Because such a check for significance is difficult with out data set, we decided for an explorative approach. We now made these limitations clearer in the manuscript.
However, our result was what we observed: we detected mutations of IQGAP2 in 3 MIBC samples compared with the earlier NMIBC samples from the same patients. The methods are clearly described.
We think a complete scientific study should contain the motivation, why we analyze this gene. We should not ignore this even it was only an inspiration, which led us to confirm the tumor suppressor role of IQGAP2 in bladder cancer, and we believe readers will consider the mutation part carefully based on our statements (e.g. weak evidence, low allele frequency) plus your criticisms. We emphasize, that the detected mutations are not the main focus of the paper but were our indicator to dive deeper into the potential role of this gene.
We revised the results part as well as the discussion.
If the editor also feels that such observations that lead to new hypothesis although labeled as not evident are not appropriate to be published in the International Journal of Molecular Sciences, we would delete that part from the manuscript and focus only on the expression analysis and functional analysis of IQGAP2.
Best regards,
Daniel Steinbach

Round 3
Reviewer 2 Report
The authors strongly believe "data is data"; while in this reviewer's many years of study on cancer genomics, the bioinformatically called variants have a pretty good chance being false positives via targeted re-sequencing, and the FP rate significantly increases when the AF is low (here they have one 0.1 and another 0.2). This concern is very inline with the TCGA's low prevalence of this gene. In fact, the low prevalence in TCGA strongly argues against the authors' claim that "will motivate further genomic researches on IQGAP2 to analyze its potential connection to the disease" because the experiment had already been done. Using a potentially flawed data to motive a study may not promote scientific rigor.
Because my critique is not concerning the remaining of the study, I will leave the decision to Editor.
Author Response
Thanks for your professional explanation and suggestions. We realized that our claim about the mutation analysis was not consistent with scientific rigor. Without re-sequencing, the results are not reliable. We will try to microdissect the samples again for resequencing as soon as possible, so we decided to delete the mutation part and focus only on expression and functional analysis of IQGAP2 in this manuscript as your advice.
Best regards,
Daniel Steinbach
